# Growth and Welfare of Rainbow Trout (*Oncorhynchus mykiss*) in Response to Graded Levels of Insect and Poultry By-Product Meals in Fishmeal-Free Diets

**DOI:** 10.3390/ani12131698

**Published:** 2022-06-30

**Authors:** Gloriana Cardinaletti, Patrizia Di Marco, Enrico Daniso, Maria Messina, Valeria Donadelli, Maria Grazia Finoia, Tommaso Petochi, Francesca Fava, Filippo Faccenda, Michela Contò, Roberto Cerri, Donatella Volpatti, Chiara Bulfon, Alberta Mandich, Alessandro Longobardi, Giovanna Marino, Lina Fernanda Pulido-Rodriguez, Giuliana Parisi, Emilio Tibaldi

**Affiliations:** 1Department of Agricultural, Food, Environmental and Animal Sciences (Di4A), University of Udine, 33100 Udine, Italy; enrico.daniso@uniud.it (E.D.); maria.messina@uniud.it (M.M.); roberto.cerri@veronesi.it (R.C.); donatella.volpatti@uniud.it (D.V.); chiara.bulfon@uniud.it (C.B.); emilio.tibaldi@uniud.it (E.T.); 2Italian National Institute for Environmental Protection and Research (ISPRA), 00144 Rome, Italy; valeria.donadelli@isprambiente.it (V.D.); mariagrazia.finoia@isprambiente.it (M.G.F.); tommaso.petochi@isprambiente.it (T.P.); alberta.mandich@gmail.com (A.M.); alessandro.longobardi@isprambiente.it (A.L.); giovanna.marino@isprambiente.it (G.M.); 3Foundation Edmund Mach, Technology Transfer Centre, San Michele all’Adige, 38010 Trento, Italy; francesca.fava@fmach.it (F.F.); filippo.faccenda@fmach.it (F.F.); 4Consiglio per la Ricerca in Agricoltura e l’Analisi dell’Economia Agraria, Centro di Ricerca di Zootecnia e Aquacoltura, Monterotondo, 00015 Roma, Italy; michela.conto@crea.gov.it; 5A.I.A.—Agricola Italiana Alimentare S.p.A. Gruppo Veronesi, 37142 Verona, Italy; 6Department of Agriculture, Food, Environment and Forestry (DAGRI), University of Firenze, 50144 Firenze, Italy; linafernanda.pulidorodriguez@unifi.it (L.F.P.-R.); giuliana.parisi@unifi.it (G.P.)

**Keywords:** alternative proteins, digestive function, immune response, *Hermetia illucens*, nutrient retention, poultry by-product meal, rainbow trout, stress, sustainable feed, welfare

## Abstract

**Simple Summary:**

Processed animal proteins, such as poultry by-product meal (PBM) and insect meal from black soldier fly (BSFM), are receiving growing interest as alternative or complementary protein sources for carnivorous farmed fish diets, due to their high nutritional value and low environmental footprint. The present study aimed to evaluate the effects of PBM and BSFM as partial substitutes for vegetable protein in fishmeal-free diets on growth, whole-body composition, nutrient-energy mass balance and retention, digestive functions, stress, metabolic status, innate immunity, and liver health in rainbow trout (*Oncorhynchus mykiss*). The overall results showed that fishmeal-free diets, including high levels of both PBM and BSFM, either singly or in combination, improved growth, dietary nutrient and energy utilization, and gut function relative to a plant-protein control diet or a fishmeal-based one, with no detrimental effects on fish welfare.

**Abstract:**

This study compared the nutrient-energy retention, digestive function, growth performance, and welfare of rainbow trout (ibw 54 g) fed isoproteic (42%), isolipidic (24%), fishmeal-free diets (CV) over 13 weeks. The diets consisted of plant-protein replacement with graded levels (10, 30, 60%) of protein from poultry by-product (PBM) and black soldier fly *H. illucens pupae* (BSFM) meals, either singly or in combination. A fishmeal-based diet was also tested (CF). Nitrogen retention improved with moderate or high levels of dietary PBM and BSFM relative to CV (*p* < 0.05). Gut brush border enzyme activity was poorly affected by the diets. Gastric chitinase was up-regulated after high BSFM feeding (*p* < 0.05). The gut peptide and amino acid transport genes were differently regulated by protein source and level. Serum cortisol was unaffected, and the changes in metabolites stayed within the physiological range. High PBM and high BSFM lowered the leukocyte respiratory burst activity and increased the lysozyme activity compared to CV (*p* < 0.05). The BSFM and PBM both significantly changed the relative percentage of lymphocytes and monocytes (*p* < 0.05). In conclusion, moderate to high PBM and BSFM inclusions in fishmeal-free diets, either singly or in combination, improved gut function and nutrient retention, resulting in better growth performance and the good welfare of the rainbow trout.

## 1. Introduction

Improving aquafeed sustainability is crucial to meeting the increasing global seafood demand and to making aquaculture a more sustainable food system that can contribute to ecological transition, as expected by the Farm to Fork strategy set by the European Green Deal [1,2,3,4]. In this context, the research on fish feed is moving towards the use of sustainable ingredients from the agri-food industry through circular bioeconomy processes [5,6].

Traditionally, the protein-source alternatives to fishmeal were mainly of vegetable origin. However, vegetable proteins contain antinutritional factors known to cause adverse physiological effects in fish [7]. More recently, processed animal proteins such as poultry by-product meal (PBM) (EU 56/2013) and black soldier fly meal (BSFM) (*Hermetia illucens*) (EU 893/2017) were employed as protein sources to replace vegetable protein because they have a low environmental footprint [8], are high in proteins with an amino acid profile almost comparable to fishmeal, and are easily available (PBM) or constantly growing in market volumes (BSFM) [9,10].

So far, fish feeding studies have mainly focused on the effect of PBM and BSFM as single fishmeal alternatives, often with inconsistent outcomes. Only a few investigations have considered both PBM and BSFM together as fishmeal replacers. We previously reported that the inclusion of graded levels of BSFM and PBM, either singly or in combination, in a vegetable-based fishmeal-free diet for rainbow trout (*Oncorhynchus mykiss*) performed comparably to or better than the fishmeal-based diet in terms of growth response [11]. Conversely, Dumas et al. [12] observed reduced growth and feed efficiency in rainbow trout in response to diets where the fishmeal protein was totally replaced by equal proportions of defatted BSFM, PBM, and a vegetable proteins blend. Improved rainbow trout gut and liver health and the down-regulation of inflammatory genes were seen after BSFM or PBM feeding when compared to the vegetable diets [11]. A similar anti-inflammatory effect at the gastrointestinal level was also observed in Gilthead seabream (*Sparus aurata*) fed fishmeal-free diets, where 20% or 40% of the crude protein from a mixture of plant proteins was replaced with crude protein from BSFM and/or PBM [13].

Different mixtures of protein sources to replace or complement the conventional ones may have an impact on fish metabolic processes. Hence, investigation of the physiological response to such dietary changes on the gastrointestinal function, blood biochemistry, and innate immune response represents a multidisciplinary approach to the optimization of novel dietary formulations. Earlier studies have investigated the expression of some gastrointestinal genes involved in protein digestion or absorption after the replacement of fishmeal protein with vegetable proteins [14,15,16,17,18,19,20]. Conversely, the effect of conventional or novel processed animal proteins, such as BSFM and PBM, on the gene expression of digestive and absorptive enzymes is poorly investigated.

The welfare of farmed fish in response to novel feed formulations is currently a key aspect of aquaculture, with consequences for sustainability and production ethics [21]. Fish welfare is commonly defined as the fish’s ability to adapt to its environment, and good welfare requires that fish be in good health, with all its biological systems working appropriately [22].

Feeding is a major risk factor for fish welfare in farming conditions [23] because sub-optimal diets can affect physiological homeostasis and the ability to cope with stressful conditions [24,25,26], with potential effects on growth performance, feed efficiency, disease resistance, and product quality [27].

Hematological and blood biochemistry parameters are commonly monitored to assess overall fish health and welfare in aquaculture studies [28,29]. Changes in these parameters are indicative of fish stress, nutritional and metabolic imbalances, altered hematological and immune responses, as well as of the health status of target organs, including the liver. Other indicators, such as the gross anatomy of the external and internal organs, the organosomatic indices, and the histopathological markers, are useful to complement the information resulting from blood screening and to provide an integrated diagnosis of fish welfare [30,31]. The concomitant monitoring of these parameters was suitable for the evaluation of the effects of fishmeal replacement with vegetable meal [32,33], insect meal [34,35,36], or a blend of plant, insect, and rendered animal proteins [11,37,38,39] on the physiology, health, and welfare of rainbow trout. In this study, these different types of parameters were integrated to assess the welfare of rainbow trout in response to novel dietary formulations.

The present investigation is part of an extensive research study, whose results on growth, gut health, microbiota composition, and fillet quality were previously reported [11,40,41]. In this paper, we present a comprehensive evaluation of the effects of feeding rainbow trout fishmeal-free diets, including graded levels of PBM and partially defatted BSFM, either singly or in combination, on whole-body composition, nutrient-energy mass balance and retention, digestive functions, and welfare.

## 2. Materials and Methods

### 2.1. Test Ingredients

The chemical composition of fishmeal (FM) and the test ingredients used in the present study are shown in Table 1. The low ash poultry by-product meal (PBM) was supplied by the ECB Company (Treviglio, Italy). It was obtained from chicken slaughterhouse leftovers cooked at 100–102 °C for 60 min and then dried at an average temperature of 110 °C for 120 min. The partially defatted black soldier fly pupae meal (BSFM) was a commercial product (ProteinX™) purchased from the Protix Company (Dongen, the Netherlands). The fishmeal used in the present study was a commercial product (Super Prime, Pesquera Diamante, San Isidro, Lima, Peru).

### 2.2. Test Diets Formulation

Eight different diets were formulated to be grossly isoproteic (42 g/100 g as fed), isolipidic (24 g/100 g as fed), and isoenergetic (22 MJ/kg). The ingredient, proximate, and amino acid composition, as well as the fatty acid profile of the test diets, are shown in Table 2 and Table 3.

A complete fishmeal-free diet, high in soybean meal, plant-protein derivatives and vegetable oil, denoted as CV, was prepared to have ratios of 10:90 and 20:80 fish to vegetable protein and fish to non-fish lipids, respectively. By contrast, a fishmeal-/fish-oil-based diet, denoted CF, was formulated with the opposite ratios of fish to vegetable protein (90:10) and lipids (80:20). These ratios were calculated considering the crude protein and lipid contribution to the whole diet of all marine-based, plant-based, and test ingredients. Six test diets were prepared by replacing graded levels of the crude protein (10, 30, and 60%) from the plant-protein-rich ingredients of the CV diet with crude protein from the insect meal *H. illucens* (H10, H30, and H60 diets) or the poultry by-product meal (P30 and P60 diets) or a combination of the two ingredients (H10P50), while maintaining the same 20:80 fish to non-fish lipid ratio, as in the CV diet.

Where necessary, the diets were supplemented with essential amino acids to meet the known nutrient requirements of rainbow trout [42]. Moreover, as shown in Table 3, and apart from obvious differences relative to the CF diet, the fatty acid profiles of the test diets were kept similar to those of the CV diet by modulating the levels of conventional vegetable oil mixture to cope with the lipid composition of the test ingredients (insect or poultry meals). The diets were manufactured by SPAROS Lda Company (Olao, Portugal) through an extrusion process in 3 and 5 mm pellets. They were stored in sealed plastic buckets in a cool and aerated room until use and while being used.

### 2.3. Test Feed Ingredients and Diet Composition Analysis

The test feed ingredients and diets were analyzed for moisture, ash, crude protein (N × 6.25), and P, according to the AOAC [43]. Their lipid content was determined following Folch et al. [44]; for the fatty acid profile, see Bruni et al. [41]. Gross energy was determined by adiabatic bomb calorimeter (IKA C7000, Werke GmbH and Co., Staufen, Germany). The chitin content of BSFM was determined according to Hahn et al. [45]. The non-protein nitrogen fraction (NPN) of the feed ingredients was determined following Careri et al. [46]. The analysis of the amino acid composition of the test ingredients and diets was performed using an HPLC system, as described by Tibaldi et al. [47]. Acid hydrolysis with HCl 6 N at 115–120 °C for 22–24 h was used for all amino acids except cysteine (Cys) and methionine (Met), for which performic acid oxidation followed by acid hydrolysis was used, as well as tryptophan, which was determined after lithium hydroxide (4 M) hydrolysis only in the ingredients. Taurine was determined only in PBM and BSFM after the water extraction and mild hydrolysis of the samples in HCl 0.1 N. The biogenic amine content of PBM and BSFM was determined according to Eerola et al. [48].

### 2.4. Fish, Rearing Conditions, and Feeding Trial Layout

The feeding trial was carried out at the aquaculture center of the Edmund Mach Institute (San Michele all’Adige, Trento, Italy) by using 1200 rainbow trout of a local strain, selected from a batch of 3000 fish hatched at the same fish farming facility. The fish were randomly distributed among 24 groups, each consisting of 50 specimens, which were stocked in 1600 L fiberglass tanks supplied with well water (temperature 13.3 ± 0.03 °C; D.O. 7.4 ± 0.5 mg/L) by a flow-through system which ensured a maximum total water volume replacement/tank/h. The fish were acclimated to the experimental condition for 2 weeks and fed a commercial diet. The fish groups (average individual weight 54.2 ± 1.45 g) were then assigned in triplicate to the 8 diets according to a completely random design.

The trial lasted 13 weeks. During the experiment, the fish were hand fed to apparent satiety, twice a day, 6 days a week. The uneaten pellets were collected, dried, weighed, and deduced from the feed amount supplied in order to obtain the feed intake.

### 2.5. Growth Performance

At the beginning and at the end of the experiment, after 24 h fasting, the initial and final fish biomasses (BWs) were recorded per each tank. The following parameters were calculated as follows:

Feed Intake (FI, g/kg/ABW/d) = feed intake per tank/[(initial biomass + final biomass)/2/days]

where ABW: average body weight (final BW + initial BW)/2;

Specific Growth Rate (SGR) = 100 × [(ln final body weight - ln initial body weight)/days];

Feed Conversion Ratio (FCR) = feed intake per tank/weight gain per tank.

#### 2.5.1. Whole-Body Composition and Nutrient-Mass Balance

Six spared fish at the start and three fish per each tank at the end of the experiment were sacrificed for whole-body composition analysis and nutrient-mass balance computation. They were minced and pooled per each tank and then freeze-dried. The samples were finely ground to powder (<500 μm) by using a centrifugal mill (ZM 1000, Retsch GmbH, Haan, Germany); for the subsequent whole-body composition analysis AOAC [44] and nutrient-mass balance, the computation was as follows:

Nutrient (energy) retention (% intake): 100 × (FBW × final whole-body nutrient content − IBW × initial whole-body nutrient content)/(nutrient intake).

Nitrogen (N) or phosphorus (P) gain (mg N or P kg^−1^ fish day^−1^): (final whole-body N or P content − initial whole-body N or P content) × ABW^−1^ × days^−1^.

#### 2.5.2. Fish Sampling, Blood Collection, and Autoptical Analysis

At the end of the feeding trial, the fish were sampled according to different protocols for biometry, autopsy, and blood and organ collection. Five fish per tank (i.e., fifteen fish for each dietary treatment; *n* = 120 fish), were rapidly anaesthetized with tricaine methanesulfonate MS222 at a dose of 100 mg/L (PHARMAQ, Fordingbridge Hampshire, UK) for blood sampling, according to Iwama et al. [49]. The fish reached a deep stage of anaesthesia within 3 min and blood was withdrawn from the caudal vein using hypodermic non-heparinized syringes. Immediately after sampling, the blood smears were prepared and the glass microcapillary tubes were filled with whole blood, for a leukocyte profile and hematocrit determination, respectively. The blood samples were allowed to clot on ice, then centrifuged at 3000× *g* for 10 min for clinical chemistry analysis and at 1000× *g* for 15 min at room temperature for immunological analysis. The serum aliquots were stored at −80 °C. The sampled fish were sacrificed with a blow to the head for autopsy and the collection of internal organs for further analysis.

The four fish subjected to the CF, CV, H60, and P60 diets were dissected for the head kidney (HK) sampling, and the tissues were maintained in HBSS without phenol red, Ca^2+^, and Mg^2+^ (Merck Life Science s.r.l., Milan, Italy) at 4 °C until the leukocyte purification. These 4 diets were selected as they were those with the maximal replacement with PBM and BSFM and were limited to the simultaneous processing/culturing of HK leukocytes from individual fish.

Small portions of liver (*n* = 6 for each dietary treatment) were fixed in Bouin’s solution for 24 h, then washed and preserved in 70% ethanol for histology.

Biometric and autoptical analyses were performed on the same fifteen fish sampled for blood collection. The general fish condition was assessed by observing the presence/absence of gross lesions on the external and internal organs. The Fulton’s condition factor (K), the hepatosomatic index (HSI), and the splenosomatic index (SSI) were calculated as follows:

K= 100 × body weight/length^3

HSI = 100 × (liver weight/body weight)

SSI = 100 × (spleen weight/body weight)

An additional six fish for each dietary treatment, after fasting for 12 h, were sacrificed with an overdose of MS 222 (300 mg/L) in order to analyze the gut gene expression and the brush border membrane (BBM) enzyme activity. This timing was chosen according to a previous study performed on rainbow trout by Borey et al. [15]. Samples of pyloric caeca (PC) anterior and posterior intestine (AI and PI, respectively) were rinsed in 0.9% NaCl and frozen in plastic tubes at −20 °C until the enzyme activity analysis. From the same fish, stomach fundus (ST), PC, and AI were also collected in RNase/DNase-free cryovials, immediately frozen in liquid nitrogen, and stored at −80 °C until the RNA extraction for gene-expression analysis.

#### 2.5.3. RT-qPCR Analysis

The expression of the genes involved in the digestive and absorptive function of the gastrointestinal tract were evaluated in the ST, PC, and AI, giving the major contribution of proximal intestine to nutrient absorption [50,51].

Approximately 60 to 80 mg of frozen tissue was disrupted, and the total RNA was extracted as previously described in Messina et al. [17]. The RNA concentration and quality (A260/280 nm ratio) were analyzed by NanoDrop™ One Microvolume UV-Vis Spectrophotometers (ThermoFisher Scientific™ Inc., Waltham, MA, USA), and the RNA integrity was assessed by standard agarose gel electrophoresis (1.2%). After extraction, the complementary DNA (cDNA) was synthesized from 1 μg of total RNA using the PrimeScript™ RT reagent Kit with gDNA Eraser (Takara Bio Europe SAS, Saint-Germain-en-Laye, France), following the manufacturer’s instructions, by using a thermocycler (BIOER LifePro Thermal Cycler; Bioer Technology Co. Ltd., Binjiang-District, Hangzhou, P.R., China). The obtained cDNA was diluted 1:10 in RNase-DNase MilliQ water and stored at −20 °C until RT-qPCR. An aliquot of cDNA was used to check the primer pair specificity.

The target genes selected and evaluated in the ST were gastric chitinase (*chia*) and pepsinogen (*peps*), while the target genes selected and evaluated in the PC and AI were oligopeptide transporter 1 (*PepT1*), neutral amino acid-transporter solute carrier (*B(0)AT1*), maltase (*malt*), and intestinal alkaline phosphatase (*iap*). Two reference genes were selected: ribosomal protein S18 RNA (*18S*) and 60S ribosomal protein L13 (*60S*). The primers for the target and housekeeping genes are listed in Table 4. They were designed based on the cDNA sequences available in the GenBank database (http://blast.ncbi.nlm.nih.gov accessed on 26 September 2018) for *Oncorhynchus mykiss* or retrieved from published sequences. The RT-qPCR was carried out in a final amount of 20 μL, containing 1 μL cDNA (50 ng), 10 μL of SsoAdvanced ™ Eva Green Supermix (Bio-Rad, Hercules, CA, USA), and 0.4 μM of each primer, using a CFX 96 real-time PCR instrutment (Bio-Rad, Milan, Italy), according to the manufacturer’s instructions. The thermal programme included 2 min at 95 °C, followed by 40 cycles at 95 °C for 10 s, 60 °C for 30 s, and an extension at 70 °C for 5 s. A negative control, containing nuclease-free water instead of the cDNA, was routinely performed for each primer set. The specificity of the reactions was verified by analysis of the melting curves (ramping rates of 0.05 °C/s over a temperature range of 55–95 °C). The relative normalized expression levels were calculated with the CFX Maestro^TM^ Software, which allowed the selection of the appropriate reference gene based on the average M value, analyzing the gene stability by means of the reference gene selection tool (CFX Maestro^TM^ Software User Guide Version 1.1; Bio-Rad, Hercules, CA, USA), using the delta-delta Ct method [52].

#### 2.5.4. Intestinal Brush Border Enzyme Activities

For enzyme analysis, the gut sections were thawed and diluted 1:10 with an iced saline buffer. The tissue samples were disrupted by Tissue Lyser II (Qiagen, Hilden, Germany) at 30 Hz for 1 min. Subsequently, each sample was centrifuged at 13,500× *g* for 10 min at 4 °C, and the supernatant was stored at −20 °C until analysis. The amount of total protein in the supernatant was determined according to Bradford [54], using Bradford reagent (Sigma-Aldrich, Milan, Italy) and bovine serum albumin (Sigma-Aldrich, Milan, Italy) as a standard. Maltase, sucrase, and intestinal alkaline phosphatase (ALP) were determined as previously described by Messina et al. [17], while the leucine aminopeptidase activity (LAP) was evaluated according to the method described by Fuentes-Quesada et al. [55]. The specific activities of the BBM enzymes were expressed as U = µmol/min/mg protein.

#### 2.5.5. Blood Chemistry and Innate Immunity Analyses

The serum cortisol (COR) concentration was detected by an automatic chemiluminometer (Immulite One, DPC Instrument Systems Division, Flanders, NJ, USA), using a chemiluminescence enzyme immunoassay (Siemens Medical Solutions-Diagnostics, USA), according to Franco-Martinez et al. [56], and osmolality by cryoscopic method (Fiske-Associates, Norwood, Massachusetts, USA), according to Cataldi et al. [57]. Selected serum biochemical parameters, including glucose (GLU), triglycerides (TAG), cholesterol (CHO), total protein (TP), albumin (ALB), urea (BUN), creatinine (CREA), aspartate aminotransferase (AST), alanine aminotransferase (ALT), and alkaline phosphatase (ALP) were analyzed by a biochemical auto analyzer (KEYVET, BPC BIOSED, Rome, Italy), using commercial kits (Giesse Diagnostics, Rome, Italy).

The hematocrit was calculated as the volume percentage of red blood cells (PCV) and determined by centrifuging micro-haematocrit heparinized capillary tubes at 12,000× *g* for 5 min in a micro hematocrit centrifuge, and then, it was measured using a roto reader [58].

The differential blood leukocyte count was further performed on blood smears fixed in methanol and then stained with May Grunwald–Giemsa. A total of 100 cells for each blood smear was counted in duplicate and identified under a light microscope according to Bulfon et al. [59].

The serum lysozyme activity was determined as previously described by Bulfon et al. [59]. The serum total myeloperoxidase activity was evaluated through the method described by Quade and Roth [60]; the method was partially modified. Fifteen microliters of serum in triplicate was added to 135 µL of HBSS in a 96-well plate and, subsequently, 100 µL/well of substrate containing 2 mM 3.3′.5.5′-tetramethylbenzidine dihydrochloride (TMB, Merck Life Science, Milan, Italy) and 6 mM of fresh hydrogen peroxide 30% (H_2_O_2_, Merck Life Science, MI, Italy). After 2 min of incubation at room temperature, the reaction was stopped by adding 50 µL/well of 2 M sulfuric acid (Merck Life Science, Milan, Italy). The optical density (OD) was read at 450 nm using a microplate reader (Sunrise, Tecan Group Ltd., Männedorf, Switzerland).

The HK leucocytes were purified, and the respiratory burst activity was quantified according to the method described by Bulfon et al. [59].

#### 2.5.6. Liver Histology

After fixation, the liver samples were dehydrated and cleared and then embedded in paraffin wax. Sections of 5 μm thickness were cut and stained with hematoxylin-eosin. The sections were then examined under an Axiophot Zeiss microscope; the images were visualized through the Zeiss Axiocam MRc5 color digital camera and acquired through the software Zeiss Application Zen 2 Blu Edition. To assess the health of the hepatic tissue, the presence/absence of the following histopathological endpoints were recorded, as described by Traversi et al. 2014 [61] and adapted for this study: hepatocyte lipid accumulation (vacuolation) (mild, moderate, and severe), nucleus peripheral position, lost of hepatic cord structure, hemorrhages, blood vessel congestion, melanomacrophage centers (MMc), granulocyte infiltration, and liver parenchyma degeneration. For each specimen (*n* = 6/dietary treatment), the endpoint occurrence was evaluated in 9 areas randomly chosen in 3 histological sections, separated by at least 50 µm each other. A grading score, representing the extension of histological alteration throughout each area, was assigned: 0 = absent; 2 ≤ 10% of field area (mild); 4 = 10–50% of field area (moderate); 6 ≥ 50% of field area (severe).

### 2.6. Statistical Analysis

Tanks were used as the experimental unit for the data on the growth performance, whole-body composition, and nutrient-mass balance, while individual fish was the experimental unit for all the remaining dependent variables. All the data were checked for normal distribution and homogeneity of variance with the Shapiro–Wilk and Levene tests, respectively. When both conditions were satisfied, a one-way ANOVA (*p*-value < 0.05) was performed to assess the effects of the diets, except in the case of the brush border membrane enzyme activities where a two-way ANOVA model (diet and intestinal tract) with interaction was adopted. When significant differences were detected, the Duncan multiple-comparison test was used for the mean comparisons.

The data on the blood chemistry parameters, differential blood leukocyte count, and innate immune response were subjected to the Kruskal–Wallis test and post hoc multiple comparisons to evaluate the effects of the dietary treatment (*p* < 0.05). Bonferroni adjustment was applied to the blood chemistry data analysis (*p* < 0.005). Discriminant analysis on the PCA factors was applied to the dataset in order to assess the discrimination among the dietary treatment groups and the associated variables. The significance of the discriminant analysis was assessed by the Monte Carlo test. The data on the histopathological parameters were analyzed by χ^2^-Test. The data analysis was carried out using the R (R Core Team, 2017), Agricolae, and Ade4 software statistical packages.

## 3. Results

### 3.1. Growth Performance

All the diets were highly palatable and were accepted from the first distribution. No mortality occurred throughout the trial. The main growth parameters, the feed and nutrient conversion, the retention efficiency, and the whole-body composition attained at the end of the experiment by the fish fed the test diets are shown in Table 5.

The fish fed the H10P50, P60, and P30 diets outperformed those fed the CF and CV diets in terms of specific growth rate (*p* < 0.05), while the diets including BSFM resulted in similar and intermediate values. The feed consumption scaled by the average body mass was slightly but significantly affected by the dietary treatments. Compared to the fish fed the CF and CV diets, which did not differ from each other, the feed intake was reduced with all the feeds, including those with medium or high levels of BSFM and PBM or with the H10P50 diet. Because of better growth and reduced feed intake, the FCR significantly improved with the latter diet when compared to that attained by the fish given CF and CV, while all the remaining treatments resulted in intermediate values between the extreme ones.

The whole-body composition was little affected by the dietary treatments in terms of the crude protein, ash, and phosphorus contents, whereas the water and fat levels resulted in a tendency towards lower and increased values, respectively, as the levels of BSFM and/or PBM increased in the diet. Extreme values for water and fat content were observed in the fish fed the CV and H60 diets (*p* < 0.05). Increased lipid content was strictly correlated (r = 0.92, *p* < 0.01) with improved gross energy retention efficiency, which was higher in the fish fed the H30, H60, and P60 diets compared to that of the fish given the CV and H10 diets, while the other treatments resulted in intermediate values.

All the diets composed by BSFM and/or PBM, irrespective of the dietary inclusion level, resulted in similarly improved (46.9 vs. 44.3%, *p* < 0.05) gross nitrogen retention efficiency when compared to the CF and CV diets, which did not differ from each other. In addition, all the test diets, including the CV one, were also similar and substantially higher in phosphorus retention relative to the CF diet (52.1 vs. 30.9%, *p* < 0.05).

### 3.2. Gene Expression

The expression of the genes involved in chitin and protein luminal digestion (*chia* and *peps*), measured in the stomachs of the fish fed the different diets, is shown in Figure 1. The *chia* was significantly upregulated (*p* < 0.05) in the fish fed the H30 and H60 diets compared to all the other dietary treatments, with the exception of H10P50. No differences were noted between the CV diet and those diets including a low level of BSFM (H10) or PBM (P30 and P60), while the lowest *chia* expression was observed in the fish fed the CF diet, which resulted in similar values compared to the trout fed CV and H10, but lower than those given the P30 and P60 diets (*p* < 0.05). The *pes* gene was also significantly upregulated, but only in the fish fed with the highest level of BSFM in the diet (H60), and no differences were observed among all the other dietary treatments.

The expression of the genes involved in the absorption of di-tripeptides (*PepT1*), amino-acid transport (*B(0)AT1*), digestion of carbohydrates (*malt*), and intestinal alkaline phosphatase (*iap*) in the pyloric caeca and anterior intestine, as affected by dietary treatment, is shown in Figure 2a,b. The expression of *PepT1* in the pyloric caeca (Figure 2a) was significantly upregulated (*p* < 0.05) in the fish fed the H10P50 diet when compared to that of the fish fed the CV, H10, and H30 diets. Intermediate values were observed with the other dietary treatments. In the proximal intestine (Figure 2b), the same gene was upregulated only in the fish fed the H10P50 diet (*p* < 0.05). In the pyloric caeca, the gene expression of the neutral amino-acid transporter (*B(0)AT1*) of the control diets did not differ. It appeared positively modulated in a nearly dose-dependent fashion by replacing the graded levels of vegetable protein from the CV diet with BSFM and/or PBM with significantly higher values at the highest substitution levels (*p* < 0.05; Figure 2a). In the anterior intestine (Figure 2b), the same gene was less affected by dietary treatments, apart from that of the fish fed the CV diet, which resulted in being significantly lower when compared to those observed with the P60 or H10P50 diets (*p* < 0.05). The gene expression of *malt* was unaffected by dietary treatment in the pyloric caeca (Figure 2a), while a significant upregulation was noted in the anterior intestine (Figure 2b) only in the fish fed the CF diet. A significant upregulation of the *iap* gene was observed in the pyloric caeca (Figure 2a) only in the fish fed the H60 and H10P50 diets (*p* < 0.05), and the same tendency was also noted also in the anterior intestine (Figure 2b).

### 3.3. Intestinal Brush Border Enzyme Activity

The specific activity of the BBME in different intestinal tracts as affected by dietary treatments is shown in Table 6. In general, the changes in enzyme activities due to the diet were relatively small in magnitude, irrespective of the intestinal tract.

A significant interaction between diet and intestinal tract was found for all BBME activities, and Table 6 presents the effects of the diet in each intestinal tract.

No diet-induced changes of maltase activity were observed in the pyloric caeca or distal intestine, while it was affected by dietary treatments in the anterior intestine, showing a significantly higher value in the fish fed the CF diet (*p* < 0.05).

Sucrase activity was significantly different across the dietary treatments in all tracts. The highest values were found in the pyloric caeca in response to the CF and P60 diets, with the former being similar to that of all the other groups, except H30 and P30 with the lowest activity. Intermediate activities were observed in the anterior intestine with the highest value again in the fish fed the CF diet (4.72 U) and the lowest in P30 (1.50 U). Conversely, the fish fed the CF diet exhibited the lowest values in the posterior intestine, though the activity in this tract was extremely low with all treatments.

The ALP activity was significantly affected by the treatments in the anterior intestine, with the CF diet again showing significantly higher activity compared to the other dietary treatments. A different situation was observed in the distal intestine, where the presence of BSFM in the diet resulted in a marginal decline in ALP activity relative to that measured in the fish fed both control diets and in those including PBM.

In the pyloric caeca and distal intestine, changes in the activity of leucine aminopeptidases due to the dietary treatment, although statistically significant, were indeed much lower, while a clearly increased LAP activity occurred in the anterior intestine with the fish fed the CV diet relative to all the other dietary treatments.

### 3.4. Fish Condition

The trout fed different diets showed general good health and no gross lesions were observed on the external and internal organs. The analysis of the condition indices (Table 7) displayed no significant differences on in Fulton’s condition factor (K) among the groups, whereas both the hepatosomatic index (HSI) and the splenosomatic index (SSI) were significantly affected by dietary treatment (*p* < 0.0001; *p* < 0.02). The HSI was significantly higher in CF compared to all the other groups, followed by P30 and P60. The HSI resulted in being similarly lower in the fish fed CV, H10, H30, and H10P50 diets, with no differences among them. The H60 diet was in an intermediate position with no difference between the latter and the P60 diet. The SSI was higher in H10 and H10P50 compared to all the other groups and significantly different relative to the CF, H30, H60, and P60 diets.

### 3.5. Blood Chemistry and Innate Immunity

Dietary treatment affected 9 out of the 13 blood chemistry parameters (Figure 3 and Figure 4) (*p* < 0.0001). Although statistically significant, the differences between the CV and the other feeding groups were relatively small in magnitude. Among the parameters regarded as primary and secondary stress indicators, cortisol (COR) did not significantly change in response to the different diets. The serum glucose (GLU) and osmolality (OSM) levels, as well as the hematocrit (HCT) values, were significantly lower in the fish fed the CV diet compared to the CF one (*p* < 0.005), whereas they were similar to those of the other dietary groups. The serum total protein (TP) content did not differ between the fish fed the CV and the CF diets, but it was reduced by the feeding of the H10 and P30 diets (*p* < 0.005). The serum albumin (ALB) concentration was significantly higher in the fish fed the CV diet compared to those given CF (*p* < 0.005) and showed a similar trend to the total protein level with the other dietary treatments, with significantly lower levels in the H10 one. No significant changes were observed in the serum urea (BUN) concentration, while the creatinine (CREA) level resulted in being significantly lower in the fish fed the CV diet and the other diets compared to CF (*p* < 0.005), except in the fish fed the H10 diet, which showed intermediate values (Figure 4). Less variations were observed in the serum lipid content. The serum triglyceride (TAG) concentration was unaffected by dietary treatment, and the cholesterol (CHO) levels were significantly higher in the fish fed the CF diet than in the other ones (*p* < 0.005). Minor changes occurred in the serum transaminases among the dietary treatments; only a significantly lower alanine aminotransferase (ALT) enzyme activity was measured in the fish fed the H60 diet compared to the H30 and H10P50 (*p* < 0.005). Alkaline phosphatase (ALP) enzymatic activity did not change in response to the dietary treatment (data not shown).

Discriminant analysis of the PCA factors, performed on the blood chemistry parameters, provided a comprehensive evaluation of the physiological status of the trout fed the different diets (Figure 5). The first two discriminant functions accounted for 61.1% of the variability of the data (the first for 33.8% and the second for 27.3%). Significant discrimination is evident between the CF diet and all the other dietary treatments (Monte Carlo test RV = 0.2 *p* = 0.001). The CF group is positioned on the X-axis and mainly depends on the cholesterol variable, which is correlated to creatinine, glucose, osmolality, and hematocrit (see circle of canonical variables). Otherwise, the CV and CV-substituted diets are aligned along the Y-axis according to a physiological gradient determined by the total protein and albumin variables. There is a substantial overlapping of the dietary groups, denoting a similar physiological status, except for the H10 and P30 groups, which were discriminated by a lower protein content, in agreement with the results of the univariate analysis.

The relative percentage of the white blood cells assessed on the blood smears was affected by the dietary treatment, with significant changes in the lymphocyte (*p* < 0.01) and monocyte (*p* < 0.05) populations (Table 8). The fish fed the H10 and H30 diets showed the highest percentage of lymphocytes, but they were significantly different only when compared to the P30 and H10P50 diets. Likewise, the same diets resulted in a significantly lower percentage of monocytes compared to the other dietary treatments, with the exception of the CV and H60, which displayed intermediate values. Despite a huge disparity in values, the fraction of blood neutrophils resulted in being highly variable within the treatments, impeding the detection of significant differences across the diets.

The innate immune parameters were measured only in the fish fed the CF, CV, H60, and P60 diets and are shown in Figure 6. They were significantly affected by the dietary treatment. A significant increase in serum lysozyme activity was observed in the fish fed the P60 diet compared to those fed the CV and CF diets (*p* < 0.05), whereas the H60 diet resulted in an intermediate value (Figure 6a). The fish given the CV diet exhibited the highest serum peroxidase activity, which did not differ from that of the fish fed the P60 diet, while significantly lower but similar levels were measured in the fish given the CF and H60 diets (*p* < 0.05) (Figure 6b). Both the H60 diet and the P60 diet resulted in the reduced respiratory burst activity of the head kidney leukocytes when compared to the CF and CV diets (*p* < 0.05), which did not differ from each other (Figure 6c).

### 3.6. Liver Histology

The livers showed normal parenchyma architecture in all the dietary groups, having regular hepatic cord structure delimited by sinusoids and patent bile ducts. Occasional signs of inflammation were observed and the presence of granulocyte infiltration and melanomacrophage centers were not affected by dietary treatment. Conversely, the diet differently induced hepatocyte lipid accumulation (χ^2^-Test *p* < 0.001), as shown in Figure 7 and Figure 8. The livers of the fish fed the CV diet were characterized by mild lipid accumulation (Figure 8A), and an increase was observed in the fish fed the diets containing BSFM and PBM (Figure 8C–E). In particular, a moderate lipid accumulation associated with sinusoid and blood vessel congestion occurred in the fish fed the P60 diet compared to the other treatments (χ^2^-Test *p* < 0.001) (Figure 8D), whereas the inclusion of insect meal in the H10P50 diet induced a small reduction in lipid accumulation in the hepatocytes (Figure 8E). In the fish fed the CF diet, the highest percentage of hepatic tissue showed severe lipid accumulation, which led to nuclei displacement at the periphery of the hepatocytes (χ^2^-Test *p* < 0.001) (Figure 8F).

## 4. Discussion

### 4.1. Growth Performance

Poultry by-product and black soldier fly meals have been extensively studied, mainly as sources of protein to replace fishmeal in the diet of different fish species [9,10,62,63,64]. The novelty of the present study was a paradigm shift in that it evaluated the response of fish to graded levels of PBM and BSFM, either singly or in combination, to replace the protein from vegetable sources in fishmeal-free diets, including substantial levels of soybean meal (SBM). It is well known that in salmonids, feeding diets high in vegetable protein sources, particularly SBM, can result in impaired growth performance and gut disorders [33,65,66,67]. This has recently also been confirmed in this research, as previously reported by Randazzo et al. [11]. In this study, the response of fish to varying dietary inclusion levels of PBM and BSFM was investigated in terms of the possible beneficial effects on nutrient retention, digestive function, and metabolic and welfare status. Relative to the CV treatment, including medium or high levels of both BSFM and PBM, either singly or in combination, led to improved nutrient and energy retention efficiency. These improvements were similar or even better than those observed in fish fed a fishmeal-based diet. A straight comparison of our results with other studies on salmonids is difficult as in most of the previous experiments the PBM and BSFM were used as fishmeal substitutes in diets including low proportions of vegetable protein sources [34,68]. In our study, PBM and BSFM were used to partially replace vegetable proteins in a basal vegetable diet. To what extent the improved growth, nutrient, and energy retention efficiency observed here reflect the improved overall nutritive value of the diets with PBM and BSFM inclusion or the concurrent shortage of vegetable protein (SBM) cannot be easily elucidated. However, as all the diets were designed to meet the rainbow trout nutrient requirements, the improvement observed in the aforementioned parameters seems conceivable, firstly, with a better overall digestible energy, amino acid supply/balance and/or improved gut health, or even improved gut microbiota composition when the plant proteins and SBM were gradually replaced by the test ingredients [40]. On the other hand, Dumas et al. [12] observed impaired growth performance, feed conversion, and a decline in the protein retention efficiency in rainbow trout of similar size in response to the increased replacement of fishmeal protein with PBM, defatted BSFM, and a blend of vegetable ingredients. These opposite outcomes could partly depend on different proportions of main alternate protein sources in the diet or in the composition of the vegetable protein blend as well as in the different protein to lipid (energy) dietary ratios between experiments. This also suggests that different ratios among major protein sources, in particular the ratio between PBM and BSFM in the diet, could be a crucial aspect that needs to be optimized in further trials to allow high growth performance to be attained even when the diets are deprived of fishmeal.

In this study, the whole-body composition of the fish was marginally affected by the dietary treatments. A few changes were observed, mostly in the water and lipid (energy) contents in the fish fed medium to high levels of BSFM, showing slightly increased body fat and, conversely, reduced water content. This is apparently in contrast with other studies where the inclusion of BSFM in the diet reduced the whole-body lipid content in rainbow trout [69] and Atlantic salmon [63]. In salmonids, lower lipid utilization and deposition in fish fed diets containing BSFM were associated with lowered lipid digestibility and with a putative adverse effect of chitin on lipid digestibility [70]. In this study, we did not measure diet digestibility; thus, we could not explore the impact of chitin on the absorption and retention of lipids and other nutrients. However, based on growth response, whole-body composition, and nutrient and energy retention efficiency, it seems that the possible adverse effect of chitin contained in BSFM diets, which ranged from 0.4% up to 1.9%, was less pronounced than in other studies. This may be related to the increased relative abundance of chitin-degrading *Actinomyces* and *Bacillus* observed in the gut microbiota of rainbow trout fed the BSFM [40].

### 4.2. Digestive Function

The host gene expression of digestive enzymes and nutrient transporters, as well as the activity of certain BBM enzymes, has been used as a complementary tool to evaluate the gut response to major dietary changes [17,20,71,72]. The stomach is involved in important physico-chemical processes for the subsequent macronutrient transit and processing, and the mucous cells at the gastric surface function as an acidic-proteolytic organ in all vertebrates [73]. It has been shown that the ability of fish to hydrolyze chitin fibers is governed by the natural feeding habits and depends on a specific set of chitinolytic enzymes, such as chitinases and chitobiase [74,75]. An important result of the present study is the finding that rainbow trout expressed the endogenous chitin hydrolysis-related gene (*chia*) in the stomach, which was upregulated in an apparent dose-dependent fashion by increasing the levels of BSFM in the diet. Moreover, the expression patterns of both *chia* and *peps* mRNAs agree with what was previously reported on other insectivorous or omnivorous animals [76,77], where the proteolytic enzyme accessibility was improved by gastric chitin-degrading enzymes [78]. The results observed here in rainbow trout fed BSFM are also consistent with previous findings on zebrafish, where an upregulation of the *chia* gene was noted in fish fed a diet where fishmeal was partially replaced by full-fat BSFM [79,80].

In the present experiment, replacing vegetable protein by moderate to high levels of PBM and/or BSFM increased the gene expression of peptide and amino acid intestinal transporters such as *PepT1* and *B(0)AT1*. Our results first confirm that the *PepT1* is abundantly expressed in the proximal intestine of rainbow trout [53,81], as also reported on marine fish species [17,18,82,83]. Secondly, these results support the finding that such a biomarker can be modulated by the source of protein in the diet [17,18,82]. The increased *PepT1* expression in the H10P50 diet, which is rich in Lys and Gly, is consistent with previous findings in trout fed diets supplemented with Lys–Gly dipeptide [52], or in carp [81] fed a diet supplemented with free lysine and glycine. In carp, this was found to be correlated with an improved feed conversion ratio and N retention. The expression and activity of *PepT1* probably depends on the availability, composition, and concentration of peptides, as well as the presence in the intestinal lumen of other components that may affect absorption dynamics [84]. According to Gilbert et al. [84], the uptake of free amino acids may be indirectly regulated by *PepT1* activity. Wenzel et al. [85] observed that the uptake of dipeptides stimulates amino acid absorption, and this could partially explain the increased expression of both *PepT1* and *B(0)AT1* in rainbow trout fed the H10P50 diet in the present study.

Intestinal alkaline phosphatase (IAP) is the major homeostatic enzyme produced by enterocytes and is known to be involved in maintaining gut health in fish and other animals [86]. Generally, the IAP activity responds to feed intake and diet composition; decreased expression is observed during starvation, and increased expression is observed after fat feeding. However, how dietary nutrients modulate both *iap* gene expression and enzyme activity is still unclear [87]. In the present study, the *iap* gene expression was upregulated in the two intestinal regions examined, particularly in the fish fed the H60 diet. This could be related to the fatty acid composition of BSFM, known to be rich in medium chain fatty acids, particularly lauric acid [88], which has been shown to increase *iap* expression and/or activity in rats and enteroid models [89,90,91]. In fact, the H60 diet was slightly higher in lauric acid than the others (Table 3), and this seems consistent with this possible explanation. Another possible reason affecting the *iap* expression evokes a role of the gut microbiota, as suggested by Bates et al. [92] and Goldberg et al. [93] and could be explained through an indirect effect of the bacterial components or metabolites on the intestinal epithelium physiology [86,87]. In our previous paper [40], we reported that rainbow trout fed the H60 diet showed high relative abundances of the *Bacillus* and *Actinomyces* genus, which are known to act as chitin degraders, thus leading to chitosan production, which in turn was found to be a stimulator of IAP activity in the gut of trout fed this polymer in the form of nanoparticles [94]. Although suggestive, this possible explanation must be considered with caution as in the present experiment there was no correspondence between the IAP gene expression and activity, probably due to the time shift between the transcription and the translation protein expression, inevitably generated by the experimental procedure.

In this study, some minor but significant changes in BBM gene expression and enzyme activity due to dietary treatments were observed only for the maltase in the anterior intestine, which appeared to be in response to the amount of dietary starch rather than the protein source. This seems consistent with the outcomes of previous studies on salmon and trout by Krogdahl et al. [95], where disaccharidase activity was found in a positive correlation with the dietary starch level. Leucine aminopeptidase (LAP) enzyme activity was used as a rough marker of the ability of fish to process dietary proteins from ingredients of different origin. In general, replacing graded levels of vegetable proteins with BSFM did not affect LAP activity along the gut apart from a numerically negligible decline with the H60 diet in the distal intestine. This is in contrast with the results of a study on Atlantic salmon [70], where a diet including 60% BSFM to replace 85% of the fishmeal protein, resulted in a marked decrease in LAP activity in the proximal and medium intestine. This adverse effect on the capability of the BBM enzyme to hydrolyze peptides into amino acids was ascribed to the chitin supplied by BSFM, which was assumed to interfere with intestinal homeostasis. Apart from the different fish species, the fish size range, the dietary levels, the composition of the plant protein, and the general culture conditions, the different outcomes of the two experiments could also depend on the higher level of dietary BSFM (chitin) in the experiment with salmon relative to the present study (i.e., 60 vs. 45%), which could have emphasized the adverse effect of chitin on intestinal mucosal homeostasis and function. In this study, the LAP activities were slightly reduced in the pyloric caeca and distal intestine when dietary plant proteins were substituted by PBM. In a previous experiment on gilthead seabream, increasing the levels of PBM in a substitution of dietary fishmeal did not affect intestinal LAP activity [96], while Hekmatpour et al. [97] found the activity of LAP to be stimulated in response to graded levels of fishmeal replacement by PBM in sobaity seabream (*Sparidentex hasta*). Beyond the different fish species, culture conditions, and major differences in the protein source being replaced (veg proteins vs. fishmeal), which could probably play a role in affecting intestinal LAP activities, there is no easy explanation to reconcile such contrasting results among the experiments. It should be stressed, however, that the magnitude of the diet-induced differences observed here in the enzyme activities was minor and were poorly reflected in terms of overall growth and protein or energy retention efficiency.

### 4.3. Fish Welfare

In this study, fish welfare was evaluated according to a function-based approach by assessing stress, metabolic, immune responses, liver health, and fish condition and by ensuring appropriate water quality and stocking density during the trial [22]. Despite its physiological consequences on welfare, stress response has been little investigated in relation to the source of dietary protein in fish. Moreover, the knowledge on the involvement of dietary protein/amino acids in modulating the primary and secondary stress response is still quite unexplored [27,98,99]. Previous studies have shown that plant-based diets induce chronic stress or reduced stress tolerance in rainbow trout [33,100,101], but little or no information is available on the stress response to diets containing insect or poultry by-product meals. In this study, the serum cortisol level did not differ among dietary treatments and was close to the basal levels reported for rainbow trout, indicating no primary stress response [102]. The osmolality and hematocrit levels, generally regarded as secondary stress indicators, were similar in the fish fed the CV and CV-substituted diets. Their values were higher in the fish fed the CF relative to those given the CV diet, probably reflecting the fact that the CF contained a large proportion of fishmeal rich in cholesterol and minerals. This also seems to be suggested by the strong correlation between cholesterol, osmolality, and hemoglobin upon the discriminant analysis (Figure 7). A close relationship between cholesterol and hematocrit has already been described in rainbow trout [58]. Nonetheless, the levels of these latter parameters measured in all the dietary groups are consistent with the normal values in rainbow trout [57,58]. In accordance with our results, osmolality was not affected in freshwater Atlantic salmon fed insect-based diets [70].

All the dietary treatments including the test ingredients, resulted in reduced serum glucose levels relative to the CF diet, with a more evident drop in the fish fed the CV diet. In this study, glycemia reflected the total carbohydrate content or levels of starch-containing ingredients in the diets. Similar to our results, no changes in plasma glucose levels were reported in rainbow trout fed diets with BSFM as a substitute for fishmeal or soybean meal [12,34,37].

The serum protein content, mainly consisting of albumin and globulins, is considered an indicator of nutritional status in fish [24,103]. In this study, the total protein and albumin concentrations were significantly lower in the fish fed the H10 and P30 diets, compared to the fish fed the CV diet, and remained in any case within the normal range of values for rainbow trout [104,105]. These results are consistent with the data reported for the sturgeon (*Acipenser schrenckii*), showing a significant decrease in serum total protein and albumin without differences in fish growth with the increasing of the dietary soy protein isolate to over 50% of the substitution of fishmeal [106]. In the sturgeon, the decline of serum proteins was ascribed to reduced protein synthesis or protein breakdown due to liver pathological conditions or intestinal mucosal inflammation/injury. This seems to be the case for rainbow trout in our experiment since, as reported by Randazzo et al. [11], histological and molecular analyses showed signs of inflammation in the distal intestine in the fish fed the H10 diet, while a moderate liver steatosis and a significant reduction in liver protein content were observed in the fish fed the P30 diet, when compared to those fed the CV diet.

We found minor changes in the serum lipid components in response to the dietary treatments. The serum triglyceride concentrations were not affected by the diets and were within normal physiological range [104]. As already noted, the fish fed the CF diet had higher serum cholesterol content, mirroring a higher cholesterol intake, while similar levels were measured in the fish fed the diets containing the test ingredients and those fed the CV diet. A concentration of around 200 mg/dL is consistent with the values reported in other studies with rainbow trout fed diets containing BSFM [34]. The fish fed the CF diet also showed higher creatinine levels, thus reflecting the high creatine content in fishmeal compared to the other ingredients.

We measured the circulating blood hepatic transaminases, AST and ALT, as biomarkers of liver tissue damage and general health condition [107]. Similar to what was previously observed in Atlantic salmon fed insect-based diets [71], in our study a high inclusion of BSFM in the H60 diet led to a lower serum ALT concentration compared to the other diets, although this difference did not reach statistical significance due to high data variability.

The results of the multivariate analysis provided a comprehensive evaluation of the physiological status of trout in response to the dietary treatments. A substantial similarity between the fish fed the CV and CV-derived diets emerged. Conversely, the H10 and P30 diets appeared to cluster separately from the other diets, and they were discriminated by lower total protein levels, denoting a physiological disturbance.

Liver health condition provides important information on fish nutritional status and welfare [108]. The inclusion of alternative protein sources in the aquafeeds was shown to affect liver histology and fatty acid composition in several fish species, depending on ingredient type and inclusion level [109,110,111,112,113]. In our analysis, several histopathological markers were used to evaluate liver health status, including lipid accumulation, circulatory disturbances, and inflammatory response [61]. The inclusion of BSFM and/or PBM in the diet did not cause severe alterations in the hepatic tissue. The fish fed the CV and H10 diets showed lower liver lipid accumulation compared to the other dietary groups, due to the well-known lipid-lowering effect of soybean meal [114]. A moderate hepatocyte lipid accumulation was observed in the fish fed the diets containing PBM and BSFM. The livers of the fish fed the P60 diet resulted in moderate lipid accumulation, which appeared slightly reduced with the diets containing BSFM alone or in combination with PBM, as already reported [11]. These findings are consistent with the HSI values and circulating transaminase levels measured in the present study (Table 7). Similar to what we observed here in the H10P50 fed trout, in *Lates calcarifer* BSFM was also associated with reduced liver lipid deposition, when included in a diet together with PBM [110]. This attenuating effect of BSFM on the liver lipid accumulation has been interpreted with regard to the higher amount of medium-chain fatty acids, in particular lauric acid, which are prone to a rapid oxidation rather than liver deposition [70,110]. Severe lipid accumulation was observed in the livers of the fish fed the CF diet (Figure 8F), which also displayed the highest HSI and serum cholesterol level compared to the other dietary groups, thus confirming our previous observations [11,41].

Despite differences in the HSI values in the function of the dietary treatment, the organosomatic indices and condition factors [30] were in line with data reported for healthy farmed rainbow trout [115].

Leukocytes are involved in fish immune response and can be modulated by abiotic and biotic stressors. Their count and formula represent useful indicators of fish health [28,116]. No effect on leukocyte formula was found in Nile tilapia (*Oreochromis niloticus*) and European sea bass (*Dicentrarchus labrax*) fed diets including BSFM at different rates of fishmeal replacement [117,118], whereas no data are available on rainbow trout fed insect or poultry by-product meal diets. In our study, the dietary treatment significantly affected the differential count of the white blood cells (Table 7). The lymphocytes resulted in being higher in the fish fed the H10 and H30 diets, whereas the monocytes and neutrophils tended to increase in the H60, P30, P60, and H10P50 diets. These results mirror the trend of increasing lysozyme activity observed in the H60 and P60 groups (Figure 6), which is probably linked to increased blood phagocytes, which are the main source of serum lysozyme [119]. The etiology of these variations remains uncertain, although it might be linked to nutrition, inflammation, or stress.

It is well recognized that the source of dietary protein affects the immune defense mechanisms in salmonids, with possible adverse effects when large proportions of fish or animal proteins are replaced in the diet by their vegetable counterparts. This is especially the case when certain soy protein derivatives such as SBM are used. In rainbow trout, several serologic and non-specific immune defense mechanisms were increased in the fish fed diets high in SBM compared to fishmeal-based diets, indicating an inflammatory or hypersensitivity reaction resulting in distal intestine pathological alterations [66,120,121,122]. Our CV diet contained a proportion of SBM at or below the level known to result in adverse effects on trout growth, gut health, and immune defense mechanisms [121,122]. When the outcomes of feeding the CV diet were compared to CF, the serum lysozyme was unaffected, but the peroxidase activity was clearly higher in the former. Peroxidases act as preventive antioxidants to detoxify peroxides that are potentially dangerous for lipids. Moreover, they function as starters to trigger oxidative reactions (stimulation of respiratory burst) activated as part of the immune defense against pathogens. Elevated peroxidase activity is likely to increase infection resistance [123]. Head kidney leukocyte respiratory burst activity was only marginally increased with the CV diet compared to the CF. This outcome does not seem to present clear evidence of an abnormal response of innate immunity in response to a diet containing soybean meal.

Among novel dietary protein sources, insects have been claimed to stimulate certain immune mechanisms in fish. However, the studies on this topic are still limited [124,125,126], and none of them evaluated the role of insect meals when included in diets high in vegetable proteins. The immune-boosting action of insects has mostly been attributed to their chitin content [125,127,128,129]. Chitin could act as stimulator of the innate immune system by binding to specific receptors, such as the macrophage mannose receptor, toll-like receptor 2 (TLR-2), interferon-γ (IFN-γ) receptor, and the Dectin-1 receptor [130]. Experiments in which insect meals obtained from *Tenebrio molitor* or *Musca domestica* were used to replace dietary fishmeal have shown increased serum lysozyme activity in rainbow trout [131] and increased phagocytic activity of peritoneal macrophages in red seabream [132]. In the present study, the fish fed a diet high in BSFM when compared to those given the CV diet resulted in lower serum peroxidase and HK leukocyte respiratory burst activities but in similar serum lysozyme values. The fish fed the BSFM diets also showed reduced HK respiratory burst activity when compared to the CF. Hence, BSFM exhibited some immunomodulatory effects but only partially confirmed previous observations, and this warrants further investigation.

Few studies are available concerning the effects of diets including PBM on the innate immune response of carnivorous fish species. In the present study, the inclusion of this ingredient in a plant-protein-rich diet (P60) resulted in elevated serum lysozyme activity compared to the values observed in trout fed CV, CF, and H60 diets. An increased serum lysozyme activity is intended as a positive indicator of non-specific humoral defense. Lysozyme is one of the main lytic factors acting against Gram-positive bacteria [133]. Rawles et al. [134] found no differences in lysozyme activity in sunshine bass fed a fishmeal-based diet or a diet including nearly 30% SBM, 30% FM, and 30% PBM. The partial replacement of fishmeal with PBM associated with BSFM (45% PBM and 10% BSFM) in the diet of juvenile barramundi (*Lates calcifer*) induced a marked increase in lysozyme activity when the fish were challenged with *Vibrio harveyi* [135]. On the other hand, a total replacement of fishmeal by PBM adversely affected the immune response, serum lysozyme activity, and stress-related genes (HSP70 and HSP90) of juvenile barramundi, whereas none of the diets had significant effects on bactericidal activity [136]. Hence, PBM inclusion in the diet, used either singly or in combination with BSFM, resulted in conflicting effects on fish immunity when the results of the present study are compared with the outcomes of other experiments. Conversely to BSFM, where chitin acts as a bioactive compound, PBM does not contain known immune-modulating compounds, making it difficult to discern its role in affecting immune response from that of other dietary components.

## 5. Conclusions

The results of this study integrate those of previous investigations which showed that moderate to high levels of PBM and BSFM, either singly or combined, to replace vegetable proteins in fishmeal-free diets, improved growth performance, gut health, and microbiota composition and reduced intestinal inflammation without impairing fillet quality [11,40,41]. The present outcomes have shown that as a consequence of better gut health and improved growth and feed efficiency, the diets which included graded levels of BSFM and PBM also led to better nitrogen retention. This was likely due to enhanced digestive/absorptive capability through an up-regulation of the genes involved in chitin/protein digestion and peptide or amino acid transport/absorption in the intestinal mucosa.

Based on a wide array of hematological, serum biochemical, and liver histological markers, as well as humoral and cellular innate immune parameters, PBM and/or BSFM have proven to be excellent complements to vegetable proteins in fishmeal-free diets by maintaining good fish welfare. The overall results provide further evidence of the suitability of these ingredients as alternative protein sources for a new generation of sustainable and healthy aquafeeds for rainbow trout.

## Figures and Tables

**Figure 1 animals-12-01698-f001:**
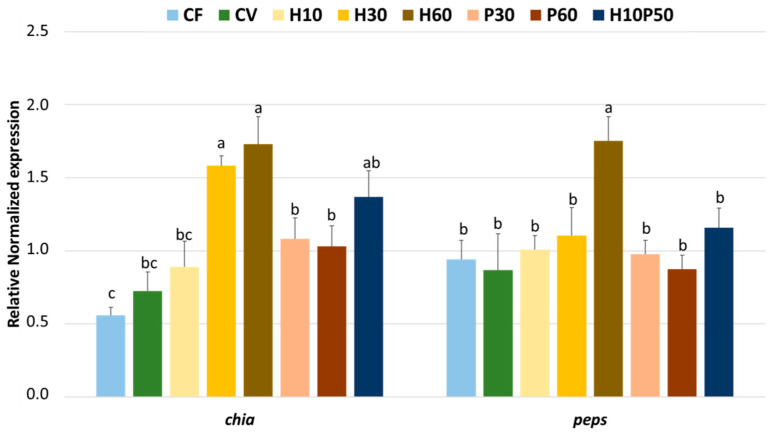
Expression of genes involved in the chitin and protein digestion (*chia* and *peps*) in the stomachs of rainbow trout fed the different diets over 13 weeks. For each gene, different superscript letters indicate significant differences among diets (mean ± error standard mean, *n* = 6) (*p* < 0.05).

**Figure 2 animals-12-01698-f002:**
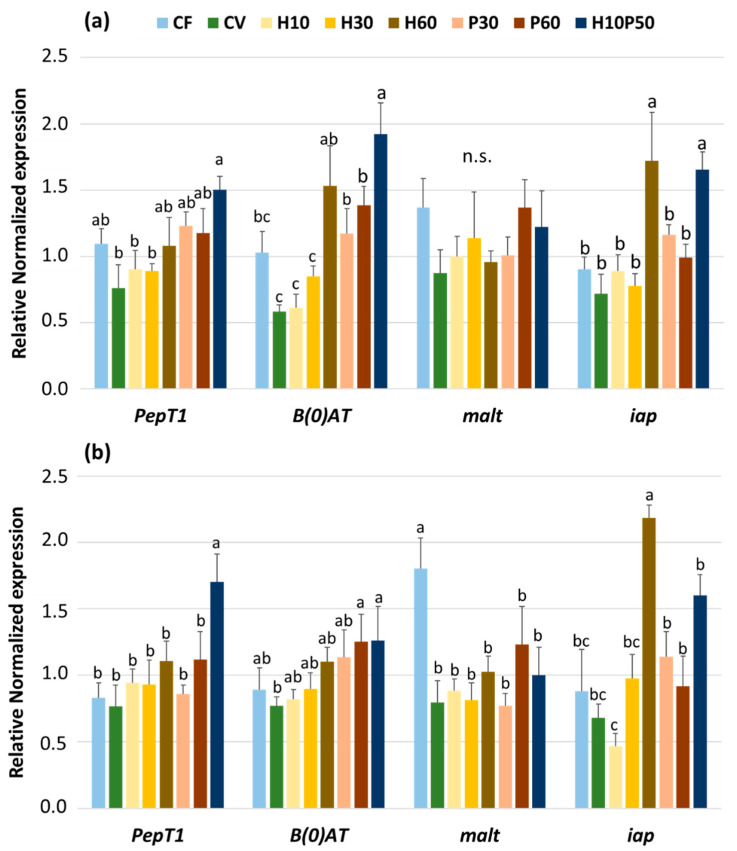
Expression of genes involved in the absorption of di-tripeptides (*PepT1*), amino-acid transport (*B(0)AT1*), carbohydrate digestion (*malt*), and intestinal alkaline phosphatase (*iap*) measured in pyloric caeca (**a**) and anterior intestine (**b**) of rainbow trout fed the different diets over 13 weeks. For each gene, different superscript letters indicate significant differences among diets (means ± esm, *n* = 6) (*p* < 0.05).

**Figure 3 animals-12-01698-f003:**
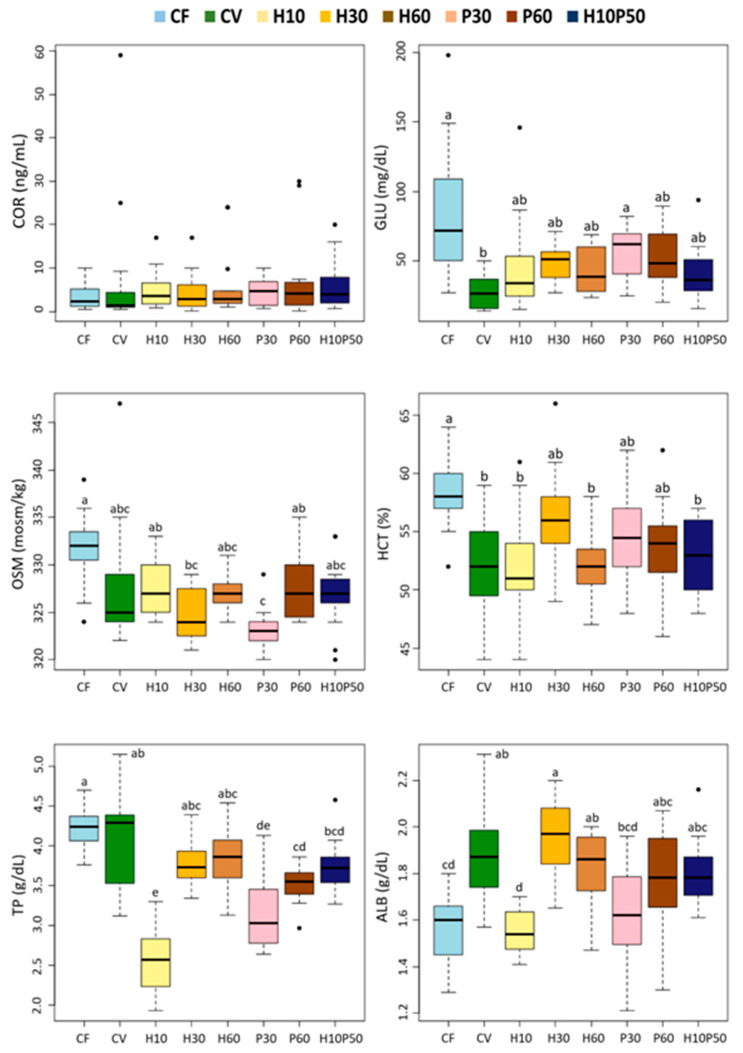
Box plots of serum biochemical parameters measured in rainbow trout fed the test diets. Different letters indicate significant differences among treatments (*p* < 0.005). Data are expressed as median, interquartile range, min., and max. values, outliers. COR: cortisol; GLU: glucose; OSM: osmolality; HCT: hematocrit; TP: total protein; ALB: albumin.

**Figure 4 animals-12-01698-f004:**
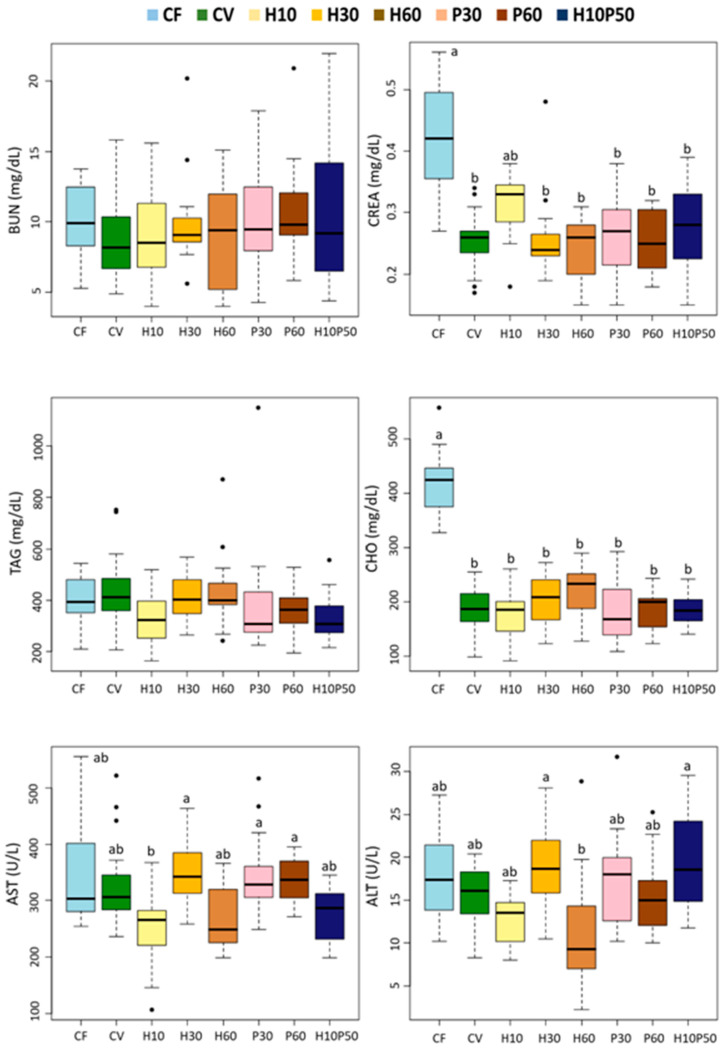
Box plots of serum biochemical parameters measured in rainbow trout fed the test diets. Different letters indicate significant differences among treatments (*p* < 0.005). Data are expressed as median, interquartile range, min., and max. values, outliers. BUN: urea; CREA: creatinine; TAG: triglycerides; CHO: cholesterol; AST: aspartate transaminase; ALT: alanine aminotransferase.

**Figure 5 animals-12-01698-f005:**
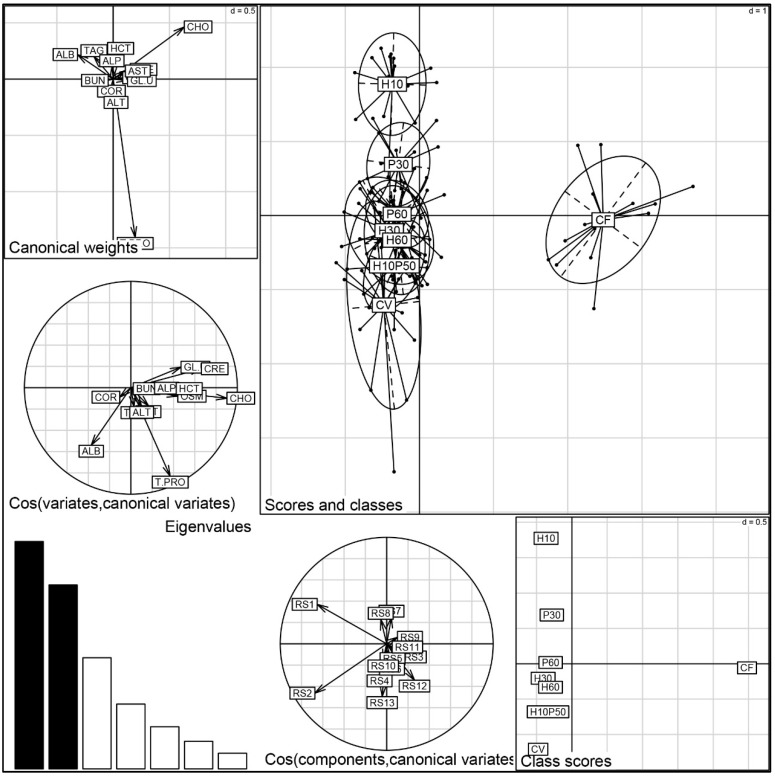
Discriminant analysis of the blood chemistry parameters measured in rainbow trout fed the test diets.

**Figure 6 animals-12-01698-f006:**
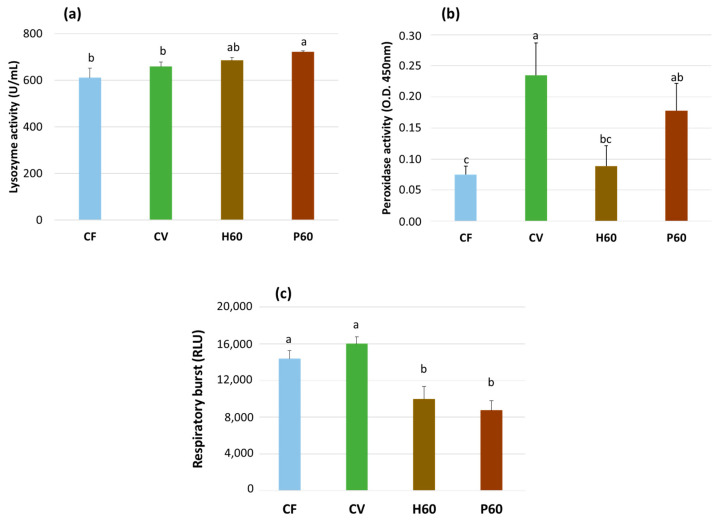
Serum lysozyme activity (U/mL) (**a**), serum peroxidase activity (O.D. 450 nm) (**b**), and respiratory burst cumulative activity (RLU/10^6^ cells/mL) of PMA stimulated head kidney (HK) leukocytes (**c**) in rainbow trout fed the test diets over 13 weeks. Data are expressed as mean ± esm (*n* = 9 for serum and *n* = 4 for HK leukocytes). Different letters indicate significant differences among test diets (*p* < 0.05).

**Figure 7 animals-12-01698-f007:**
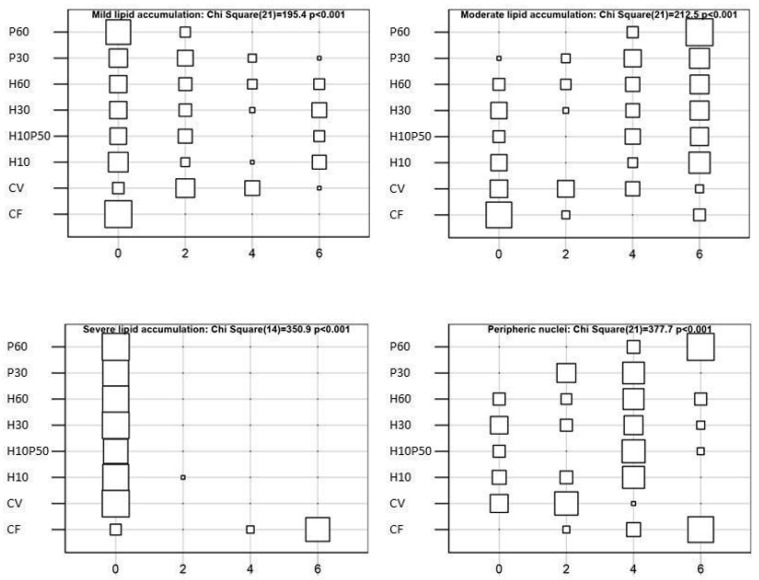
Representation of the contingency tables displaying the frequency of hepatic lipid accumulation and nucleus displacement observed in livers of rainbow trout fed the test diets. Histological alterations were evaluated in 9 areas randomly chosen on the basis of their percentage of occurrence and scored as follows: 0 = absent; 2 ≤ 10% of field area (mild); 4 = 10–50% of field area (moderate); 6 ≥ 50% of field area (severe).

**Figure 8 animals-12-01698-f008:**
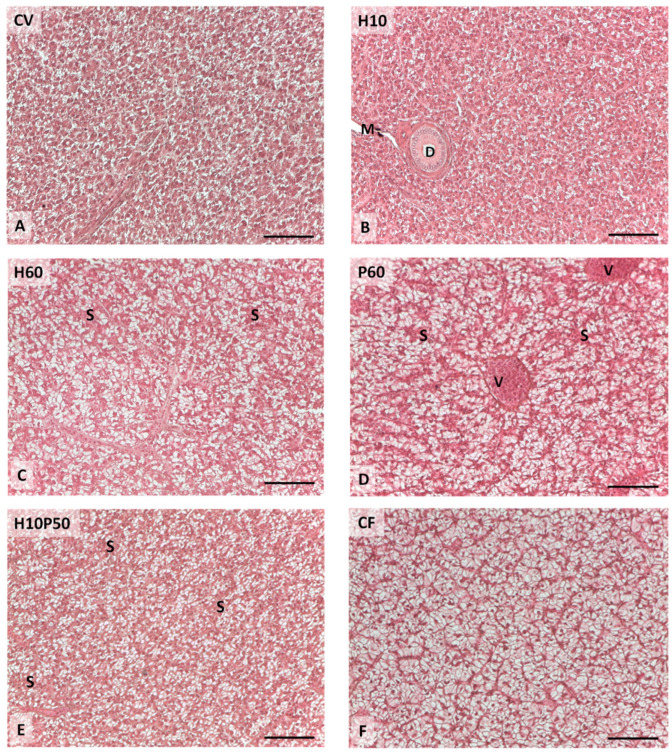
Liver histological micrographs of the rainbow trout fed the test diets showing a different lipid accumulation degree: mild (**A**,**B**); moderate (**C**–**E**); severe (**F**). Legend: D: bile duct; M: melanomacrophages; S: sinusoid; V: blood vessel. Hematoxylin-eosin staining. Scale bar = 50 μm.

**Table 1 animals-12-01698-t001:** Chemical composition of the fishmeal and test ingredients.

	FM	PBM	BSFM
Dry Matter %	95.7	94.2	95.6
N × 6.25 %	65.3	65.6	53.1
Fat %	11.5	14.8	20.8
Ash %	17.5	12.4	6.4
Carbohydrate ^1^ %	-	1.4	15.3
Chitin g/100 g	-	-	4.7
NPN ^2^ g/100 g	n.d.	3.0	1.9
Gross Energy MJ/kg	19.2	21.3	20.8
**EAA ^3^ g/100 g**			
Arg	4.8	4.8	2.6
His	2.0	1.2	1.4
Ile	3.0	1.7	1.6
Leu	4.7	4.0	3.6
Lys	5.3	3.1	2.8
Met + Cys	2.5	1.7	1.5
Phe	3.1	2.1	1.9
Tyr	2.5	1.7	3.3
Thr	3.1	2.1	2.1
Trp	0.5	0.8	0.4
Val	3.4	2.9	3.3
**NEAA ^4^ g/100 g**			
Ala	3.8	5.0	4.1
Asp	5.7	6.8	6.2
Glu	8.2	11.3	6.9
Gly	4.0	6.4	2.9
Pro	2.9	5.1	4.0
Ser	3.1	2.9	2.6
Taurine mg/kg	376	1536	39
**Biogenic amines mg/kg**			
Tryptamine	2	9	7
2-PHE	5	4	6
Putrescine	69	36	46
Cadaverine	166	164	18
Histamine	134	18	12
Tyramine	42	48	3
Spermidine	23	75	133
Spermine	12	62	18

^1^ Calculated by difference as 100—(Water + CP + Lipid + Ash); ^2^ non-protein nitrogen; ^3^ EAA, essential amino acid (including Cys and Tyr); ^4^ NEAA, non-essential amino acids; n.d., not determined.

**Table 2 animals-12-01698-t002:** Ingredient composition of the test diets.

	Diets							
Ingredient Composition %	CF	CV	H10	H30	H60	P30	P60	H10P50
Fishmeal ^1^	47.5	-	-	-	-	-	-	-
CPSP 90 ^2^	5.0	5.0	5.0	5.0	5.0	5.0	5.0	5.0
Soybean meal	-	23.0	20.4	16.0	9.0	16.0	9.0	9.0
Protein-rich veg. mix ^3^	-	31.4	27.2	19.4	7.8	18.7	6.0	6.3
Rapeseed meal	3.8	3.5	3.2	2.5	2.4	2.5	2.0	2.0
Hermetia meal ^4^	-	-	7.8	22.7	45.0	-	-	7.8
Poultry by-product meal ^5^	-	-	-	-	-	17.8	36.0	29.7
Whole wheat	15.6	-	-	2.8	6.2	9.9	18.6	14.5
Pea meal	7.0	7.1	9.2	6.8	3.0	6.9	3.0	3.5
Fish oil	15.1	4.4	4.4	4.4	4.4	4.4	4.4	4.4
Vegetable oil mix ^6^	4.3	17.7	16.7	14.8	12.0	15.5	13.4	13.2
Vit. and min. premix ^7^	0.2	0.2	0.2	0.2	0.2	0.2	0.2	0.2
Dicalcium Phosphate	-	3.0	3.0	2.8	2.7	0.6	-	1.8
Betaine HCl	-	1.5	-	-	-	-	-	-
L-Lysine	-	1.2	0.9	0.7	0.5	0.6	0.6	0.8
DL-Methionine		0.45	0.45	0.40	0.35	0.35	0.25	0.25
L-Tryptophan		0.05	0.02			0.04	0.05	0.03
Celite	1.5	1.5	1.5	1.5	1.5	1.5	1.5	1.5

^1^ Super Prime fishmeal, Pesquera Diamante, San Isidro, Lima, Peru. ^2^ CPSP90, fish protein concentrate, Sopropeche, Boulogne sur mer, France. ^3^ Soy protein concentrate (Soycomil) and wheat gluten 1:1 w/w. ^4^ ProteinX™, Protix, Dongen, the Netherlands. ^5^ Poultry by-product meal low ash, ECB Company S.r.l. Treviglio (BG), Italy. ^6^ Composition %: rapeseed oil, 50; linseed oil, 40%; palm oil, 10%. ^7^ G supplying per kg of supplement: Vit. A, 4,000,000 IU; Vit D3, 850,000 IU; Vit. K3, 5000 mg; Vit.B1, 4000 mg; Vit. B2, 10,000 mg; Vit B3, 15,000 mg; Vit. B5, 35,000 mg; Vit B6, 5000 mg, Vit. B9, 3000 mg; Vit. B12, 50 mg; Vit. C. 40,000 mg; Biotin, 350 mg; Choline, 600 mg; Inositol, 150,000 mg; Ca, 77,000 mg; Mg. 20,000 mg; Cu, 2500 mg; Fe, 30,000 mg; I, 750 mg; Mn, 10,000 mg; Se, 80 mg; Zn, 10,000 mg.

**Table 3 animals-12-01698-t003:** Chemical and amino acid composition and fatty acid profile of the test diets.

	Diets							
	CF	CV	H10	H30	H60	P30	P60	H10P50
**Proximate composition % as fed**								
Dry matter	92.4	91.2	90.5	91.2	91.1	90.7	94.0	92.9
Crude protein	42.0	42.1	41.9	41.5	42.0	41.8	42.2	41.9
Crude lipid	23.9	23.9	24.2	23.8	24.1	23.9	24.0	24.2
Ash	9.5	8.0	8.2	8.3	8.6	6.7	6.8	8.4
Carbohydrate ^1^	17	17.2	16.2	17.6	16.4	18.3	21	18.4
Total P	1.37	0.70	0.73	0.75	0.77	0.78	0.80	0.78
Gross energy (MJ/kg)	22.4	21.9	22.5	21.9	22.5	22.5	22.9	22.9
**Amino acid composition % as fed**								
Arg	2.4	2.6	2.6	2.5	2.3	2.7	2.8	2.7
His	0.9	1.0	1.0	1.0	1.1	0.9	0.9	0.9
Ile	1.6	1.7	1.7	1.7	1.7	1.7	1.6	1.6
Leu	2.6	2.9	2.9	2.9	2.8	2.9	2.9	2.8
Lys	2.9	2.9	2.8	2.8	2.8	2.7	3.0	3.1
Met	1.1	1.0	1.0	1.0	1.1	1.0	1.0	1.0
Cys	0.3	0.6	0.6	0.5	0.5	0.6	0.6	0.5
Phe	1.8	1.9	1.9	1.8	1.8	1.8	1.7	1.7
Tyr	1.3	1.2	1.5	1.9	2.3	1.3	1.3	1.5
Thr	1.6	1.4	1.5	1.5	1.6	1.5	1.6	1.6
Trp	0.5	0.4	0.4	0.5	0.5	0.4	0.4	0.4
Val	1.9	1.8	1.9	1.9	2.1	1.9	2.0	2.0
Asp	3.0	3.0	3.1	3.2	3.4	3.1	3.3	3.3
Glu	4.9	8.7	8.1	7.1	5.6	7.6	6.4	6.2
Ala	2.1	1.5	1.7	2.0	2.4	1.9	2.3	2.3
Gly	2.5	1.8	1.9	2.0	2.1	2.7	3.3	3.2
Pro	1.7	2.9	2.8	2.7	2.6	2.8	2.7	2.6
Ser	1.8	1.9	1.9	1.8	1.8	1.8	1.7	1.7
**Fatty acid profile (% FAMEs)**								
C12:0	0.1	0.1	0.5	1.2	2.1	0.1	0.2	0.8
C14:0	6.0	1.7	1.8	1.9	1.7	1.7	2.0	1.8
C16:0	17.0	11.6	12.2	11.7	11.9	11.7	11.7	12.0
C18:0	3.6	3.5	3.4	3.4	3.6	3.5	3.4	3.6
C18:1n-9	24.5	34.0	33.4	33.1	34.1	33.8	32.7	33.7
C18:2n-6, LNA	8.7	17.3	17.2	17.5	17.4	17.4	17.3	17.7
C18:3n-3, ALA	2.8	19.0	18.5	18.6	18.7	18.9	18.4	18.3
C20:5n-3, EPA	11.1	3.1	3.2	3.0	2.9	3.1	3.0	2.7
C22:6n-3, DHA	6.2	1.6	1.7	1. 6	1.5	1.6	1.6	1.4
ΣSFA	28.0	17.8	18.8	19.0	18.2	17.9	20.0	19.0
ΣMUFA	37.5	39.1	38.3	38.1	39.2	38. 9	37.5	38.8
ΣPUFAn-6	10.2	17.6	17.6	17.9	17.8	17.8	17.6	18.1
ΣPUFAn-3	20.8	24.6	24.3	24.1	24.0	24.5	23.9	23.2

^1^ Calculated by difference as 100—(Water + CP + Lipid + Ash). FAME: fatty acid methyl ester; LNA: linoleic acid; ALA: alpha-linolenic acid; EPA: eicosapentaenoic acid; DHA: docosahexaenoic acid; SFA: saturated fatty acids; MUFA: monounsaturated fatty acids; PUFA: polyunsaturated fatty acids. The rainbow trout EEA requirements (NCR, 2011) (g/100 g diet): Arg 1.5; His 0.8; Ile 1.1; Leu 1.5; Lys 2.4; Met + Cys 1.1; Phe 0.9; Phe + Tyr 1.8); Thr 1.1; Trp 0.3; Val 1.2.

**Table 4 animals-12-01698-t004:** Primers used to evaluate gene expression by RT-qPCR.

Gene	Accession Number	Primer Forward (5′-3′)	Primer Reverse (5′-3′)	Ref.
*chia*	EU877960	CGTTCATCAGCAGCGTTATCA	CAGCATCAGACGACGAGGAAGGT	-
*peps*	EU880230	TGTCCGAGTGTAATGTCAAG	CCATAGGTTTGTAGGGGAAC	-
*malt*	XM_036961527	ATACTGCCCTGATTGGAC	TATTCCTGCTGCTCTCATTT	-
*PepT1*	KY775396	CCTGTCAATCAACGCTGGT	CACTGCCCATAATGAACACG	[53]
*B(0)AT1*	KY775397	ACCTCAAAACCTGCGACTTG	CCACCGTTCCTTCTATGCTG	-
*iap*	XM_021609753	CACAGCCCCCTTATCTCCTT	TCACCAACGCTCAAAACACT	-
*18S*	FJ710874	GCAAGTCTAAGTACACACG	CGAAGTTATCTAGAGTCACC	[11]
*60S*	XM_021601278	TTCCTGTCACGACATACAAAG	GTAAGCAGAAATTGCACCATC	[11]

**Table 5 animals-12-01698-t005:** Growth performance, whole-body composition, nutrient gain, and retention in rainbow trout fed the test diets over 13 weeks.

	Diets								
	CF	CV	H10	H30	H60	P30	P60	H10P50	Pooled s.e.
Final weight (g/fish)	231.2 ^b^	227.9 ^b^	235.0 ^ab^	239.1 ^ab^	241.0 ^ab^	240.0 ^ab^	244.0 ^ab^	254.8 ^a^	1.92
Feed intake (g/kg ABW/d)	10.8 ^b^	11.0 ^a^	11.0 ^a^	10.7 ^b^	10.6 ^bc^	10.7 ^b^	10.7 ^b^	10.5 ^c^	0.08
SGR	1.61 ^cd^	1.57 ^d^	1.63 ^bc^	1.63 ^bc^	1.63 ^bc^	1.64 ^abc^	1.66 ^ab^	1.69 ^a^	0.007
FCR	0.78 ^abc^	0.80 ^a^	0.79 ^ab^	0.76 ^bcd^	0.76 ^bcd^	0.76 ^bcd^	0.75 ^cd^	0.73 ^d^	0.004
**Whole-body composition**									
Water %	72.1 ^ab^	72.5 ^a^	71.7 ^b^	71.4 ^b^	71.3 ^b^	71.4 ^b^	71.3 ^b^	71.8 ^b^	0.33
Protein %	14.0	14.5	14.8	14.4	14.1	14.4	14.5	14.2	0.27
Fat %	11.3 ^b^	10.3 ^c^	10.9 ^bc^	12.0 ^ab^	12.6 ^a^	11.6 ^b^	11.9 ^ab^	12.2 ^ab^	0.51
Ash %	2.3	2.3	2.4	2.3	2.2	2.4	2.4	2.4	0.18
Phosphorus %	0.34	0.32	0.33	0.32	0.37	0.33	0.35	0.35	0.021
Energy kJ·g^−1^	7.86 ^b^	7.55 ^c^	7.65 ^c^	8.08 ^ab^	8.13 ^a^	7.82 ^bc^	7.97 ^ab^	7.86 ^b^	0.155
**Daily gain**									
Nitrogen (mg/kg ABW)	304 ^c^	313 ^bc^	325 ^a^	318 ^ab^	311 ^bc^	320 ^ab^	326 ^a^	320 ^ab^	7.3
Phosphorus (mg/kg ABW)	45.4	41.3	43.5	41.7	47.8	44.4	49.0	49.2	4.18
**Retention % of intake**									
Nitrogen	44.2 ^b^	44.3 ^b^	46.5 ^a^	47.6 ^a^	46.4 ^a^	47.6 ^a^	46.9 ^a^	46.4 ^a^	1.10
Phosphorus	31.2 ^b^	54.7 ^a^	54.9 ^a^	52.7 ^a^	59.2 ^a^	53.6 ^a^	58.0 ^a^	58.3 ^a^	5.24
Energy	49.7 ^bc^	46.0 ^c^	47.3 ^c^	52.3 ^a^	52.9 ^a^	50.2 ^b^	52.1 ^ab^	50.1 ^b^	1.43

Row means with different superscript letters denote significant differences among diets (a, b, c, d; *p* < 0.05).

**Table 6 animals-12-01698-t006:** Specific activity (U) of maltase, sucrase, alkaline phosphatase (ALP), and leucine aminopeptidases (LAP) in different rainbow trout intestinal tracts (*n* = 6).

	Diets								
	CF	CV	H10	H30	H60	P30	P60	H10P50	Pooled s.e.
**Pyloric Caeca**									
Maltase	47.03	48.42	41.58	41.28	57.79	34.14	50.68	43.17	2.523
Sucrase	5.42 ^ab^	4.49 ^abc^	3.23 ^bc^	2.93 ^c^	4.54 ^abc^	2.38 ^c^	6.17 ^a^	3.43 ^bc^	0.461
ALP	0.69	0.74	0.67	0.99	0.61	0.60	0.46	0.68	0.054
LAP	2.91 ^abc^	3.08 ^ab^	2.21 ^bc^	3.71 ^a^	1.98 ^bc^	2.07 ^bc^	1.67 ^c^	2.50 ^abc^	0.238
**Anterior intestine**									
Maltase	40.19 ^a^	24.03 ^b^	19.39 ^b^	19.63 ^b^	22.97 ^b^	13.76 ^b^	21.74 ^b^	18.70 ^b^	2.755
Sucrase	4.72 ^a^	2.05 ^cd^	2.33 ^cd^	2.46 ^bcd^	3.64 ^ab^	1.50 ^d^	2.72 ^bc^	2.74 ^bc^	0.035
ALP	0.68 ^a^	0.27 ^bc^	0.29 ^bc^	0.25 ^bc^	0.42 ^b^	0.30 ^bc^	0.39 ^bc^	0.22 ^c^	0.053
LAP	1.37 ^a^	0.97 ^b^	0.86 ^b^	0.83 ^b^	0.82 ^b^	0.62 ^b^	0.97 ^b^	0.70 ^b^	0.081
**Distal intestine**									
Maltase	5.02	5.29	5.44	4.62	6.07	4.90	7.40	3.96	0.366
Sucrase	0.49 ^b^	0.78 ^ab^	0.90 ^a^	0.87 ^ab^	1.15 ^a^	0.99 ^a^	0.82 ^ab^	0.77 ^ab^	0.068
ALP	0.12 ^ab^	0.12 ^ab^	0.06 ^c^	0.06 ^c^	0.09 ^bc^	0.11 ^abc^	0.15 ^a^	0.10 ^abc^	0.010
LAP	0.22 ^c^	0.34 ^a^	0.30 ^ab^	0.30 ^ab^	0.26 ^bc^	0.26 ^bc^	0.25 ^bc^	0.21 ^c^	0.016

Within the intestinal tract, row mean values not sharing the same superscript letters differ significantly: a, b, c, d; *p* < 0.05.

**Table 7 animals-12-01698-t007:** Condition indices in rainbow trout fed the test diets over 13 weeks. Data are expressed as mean ± esm. Different letters indicate significant differences among groups (*p* < 0.05). K: condition factor; HSI: hepatosomatic index; SSI: splenosomatic index.

	Condition Indices		
Diets	K	HSI	SSI
CF	1.56 ± 0.02	1.90 ± 0.07 ^a^	0.10 ± 0.01 ^c^
CV	1.62 ± 0.02	1.28 ± 0.03 ^e^	0.13 ± 0.01 ^ab^
H10	1.64 ± 0.03	1.26 ± 0.05 ^e^	0.14 ± 0.01 ^a^
H30	1.61 ± 0.03	1.27 ± 0.04 ^e^	0.11 ± 0.01 ^bc^
H60	1.59 ± 0.03	1.30 ± 0.05 ^de^	0.11 ± 0.01 ^bc^
P30	1.61 ± 0.03	1.48 ± 0.06 ^bc^	0.13 ± 0.01 ^ab^
P60	1.60 ± 0.02	1.44 ± 0.05 ^cd^	0.12 ± 0.01 ^bc^
H10P50	1.62 ± 0.03	1.22 ± 0.03 ^e^	0.14 ± 0.01 ^a^
K-W Test	ns	*p* < 0.0001	*p* < 0.02

**Table 8 animals-12-01698-t008:** Relative percentage of white blood cells (WBC) in rainbow trout fed the test diets over 13 weeks. Different letters indicate significant differences among groups (*p* < 0.05).

	Diets								
	CF	CV	H10	H30	H60	P30	P60	H10P50	K-W Test
%WBC									
Lymphocytes	96.2 ^bcd^	96.7 ^abc^	97.0 ^ab^	97.5 ^a^	96.3 ^bcd^	94.7 ^d^	96.0 ^bcd^	94.4 ^cd^	*p* < 0.01
Neutrophils	1.8	1.7	1.9	1.4	2.0	2.4	2.0	3.1	ns
Monocytes	2.0 ^a^	1.7 ^ab^	1.1 ^b^	1.1 ^b^	1.8 ^ab^	2.9 ^a^	2.0 ^a^	2.5 ^a^	*p* < 0.05

## Data Availability

Not applicable.

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
