# Peer review of "Growth and Welfare of Rainbow Trout (Oncorhynchus mykiss) in Response to Graded Levels of Insect and Poultry By-Product Meals in Fishmeal-Free Diets"

_animals, 2022, doi:10.3390/ani12131698_

Round 1

Reviewer 1 Report

The present manuscript investigated the effects of feeding rainbow trout fishmeal-free diets including graded levels of PBM and partially defatted BSFM singly or in combination, on whole body composition, nutrient-energy mass balance and retention, digestive functions, welfare and innate immune response. The design is good, and a lot of information were obtained, which would be helpful for the application of PBM and BSFM in aquafeed. On the other hand, there are some aspects which needs to be addressed.

1.     Abstract, please put “p<0.05” inside the sentences when the difference is significant;

2.     It is necessary to define the concept “welfare” and make a distinction between it and health status;

3.     It is mentioned that PBM and BSFM could be applied as alternative or complementary protein sources, while fat content in PBM and BSFM are not low, and the fatty acid profile are not available, the related influence should be discussed more sufficiently.

4.     Discussion part, it is suggested to divided into several sections which would be more easier for the readers to understand the various aspects.; The plant sources applied in the experimental diet should be discussed.;

5.     Title of 3.4, the description “Fish health conditions” is not clear; HSI was found to be the highest in FM group, why? It would be better if the data of liver composition are available. Also, it would be better if the liver histological picture for observation could be provided in 3.6.

Reviewer 2 Report

The study by Cardinaletti et al. demonstrated the applicability of two different sustainable alternative protein sources in the diets for rainbow trout, as a replacement for vegetable protein sources, which have been shown to negatively affect  fish physiology. This study is of great interest because data about the effects of sustainable ingredients used in combination are scarce.

The manuscript is very well prepared, the information provided is new and the justification of the investigation is well defined. The findings are presented in a detailed and comprehensive way in the results section. The authors discuss the results point by point in the discussion section and explain/attempt to explain the observed effects of the alternative ingredients on the parameters tested. The references list is appropriate and well suited within the Introduction or Discussion part.

My only suggestion to the authors is to elaborate a little more on the methodology.

Reviewer 3 Report

The authors investigated the effect of growth and welfare of rainbow trout (Oncorhynchus mykiss) in response to graded levels of insect and poultry by-product meals in fishmeal-free diets. They designed eight treatments to test their hypothesis. This manuscript (MS) was clearly written and easy to understand. This work can help the sustainability of this species farming as few research on this topic has been done. However, some major issues significantly compromised the quality of this MS.

Major comments:

  • First, the manuscript needs to be edited by a native English speaker to improve the language of the MS and fix errors. I corrected some of them, but still, more work needed to be done.
  • The profile of fatty acids of at least diets is required to make a solid conclusion. The main problem with these ingredients is lipid profile and not amino acids. Please provide it in the next version of the MS.

However, I have touched on some more points that can contribute to the improvement of this MS.

Minor comments

  • Line 42-48, it is too general, please explain with more details.
  • Line 32-37, please revise the English of this part.
  • Line 38-48, please revise it.
  • Line 39 and 51, using abbreviations for the name of fish species is not common and correct. Please revise it throughout the MS.
  • Line 73-76, is not clear, please revise it.
  • Line 89-101, please revise it.
  • Line 102, the correct term is “blood biochemistry”
  • Line 103, the appropriate review papers should be cited. I suggest you cite this paper: https://doi.org/10.3390/biology10121236

  • Line 106-111, please revise.
  • Line 113, is not clear what were your previous studies and what is the novelty of the present work. Please revise this section and address the mentioned comment.
  • Table 1, please add the chemical composition, amino acid profile, and biogenic amines of fish meal.
  • Line 131-135, please revise it.
  • Line 136-138, is so confusing, please revise it.
  • Line 142-150, again, please revise it. It is so confusing. Further, please introduce better terms for the name of treatments.
  • CV and CF are the common names for the formula of growth. Please change the name of treatments to something better.
  • The main problem related to these ingredients is fatty acid profile. Therefore, the fatty acids profile of ingredients and preferably fish is required to make a solid conclusion. Otherwise, with only the amino acid profiles, we cannot come to decision.
  • Please add the amino acid requirements of rainbow trout to Table 3. Further, 47% protein_15 fish oil is too much for this fish species.
  • Line 349-366, I did not understand why you did not analyse data with simple ANOVA. More explanation is required to answer the concern regarding the applied statistical method.
  • Table 5, please add the initial weight.
  • Further, SE of 1.9 is too few for fish of this size and does not make sense.
  • Please make sure the topics you mentioned in the introduction are matched with other parts of MS.
  • Line 390 and throughout the MS, if there is no significant difference, no need to report P-value.
  • Figure 3, Table 8 and others, too many abbreviations in the MS and is hard to read. Please revise the whole MS from this point.
  • Line 526, what was the aim of doing PCA? Please clarify it.
  • In the discussion section, please add more concepts regarding comparing your study with your previous works. You have a chance to make a solid conclusion from all your data.
  • Here and elsewhere, report P uppercase and italic (P<0.05).
  • Please reorder the keywords alphabetically and capitalize each word.
  • Here and throughout the MS, please first mention the common name plus scientific name, and for the rest of the MS, just report the common name.
  • Some parts of the discussion are better updated with research in 2021 and 2022 as they refer to some old references. Please update the discussion with the latest studies as much as possible.
  • Although you wrote this section well, you can still improve it by answering these questions and annotating them in the discussion section. Why were these results observed? Discuss more possible reasons.
  • The conclusion needs to be revised and add more comprehensive concepts there.

When revising your manuscript, please consider all issues mentioned in the reviewers' comments carefully with clear outlines for every change made in response to their comments including suitable rebuttals for any comments you deem inappropriate. Please itemize your response to each review comment, and highlight the revised at re-submission.

Kind regards

Round 2

Reviewer 3 Report

The authors improved the quality of the MS. However, there is a big issue with the fatty acids profile of the diets. CF has 47.5% fish meal and 15.6% fish oil compared to other diets without fish meal and 4% fish oil. When we look at the n-3, CF treatment is lower than other diets. It seems an error occurred here. 

Author Response

The reviewer takes a point that may at first glance be erroneous, but it is instead logical and correct.

In fact, as expected and evident from Table 3, the CF diet had a much higher EPA and DHA content than other diets and this reflects the high content of fishmeal and fish oil compared to other diets.

Therefore, the slightly higher proportion of total n-3 in diets other than CF, is easily explained by considering the high proportion of ALA (C18:3n-3) in the diets themselves relative to CF, as it is clearly indicated in Table 3, which in turn reflects the high proportion of vegetable oils rich in ALA (rapeseed oil; linseed oil) used in the oil mixture whose composition has been clearly indicated in footnote 6 of Table 3. 
